# FOE-RL: FLEXIBLE ONLINE REINFORCEMENT LEARNING FOR EFFICIENT INFERENCE IN LARGE REASONING MODELS

## ABSTRACT

Recent advancements in large reasoning models have significantly enhanced their reasoning abilities. However, recent studies have shown that these models often experience "overthinking," even when handling relatively simple questions. In this paper, we propose a flexible online reinforcement learning method that estimates the difficulty of a problem in real-time and predicts an appropriate output length. Based on this, we design a length reward function and a flexible reward trend monitor, which dynamically activates or deactivates the length reward according to smoothed correctness rewards. Experimental results demonstrate the effectiveness of our approach. Compared to training methods that rely solely on correctness rewards, our approach significantly improves model accuracy while substantially reducing the average response length. On the MATH dataset, our method reduces the output token count by over 40% and increases accuracy by more than 4%. Across multiple testing benchmarks, it maintains or even enhances model performance while consistently lowering token usage. Furthermore, we observe that the method exhibits a self-regulating output length capability: depending on the model's own capacity and question difficulty, it automatically converges toward an optimal output length range, achieving higher accuracy in the process.

## 1 INTRODUCTION

In recent years, large reasoning models (LRMs) like OpenAI's o1 (Jaech et al., 2024) and DeepSeek's R1 (Guo et al., 2025) have demonstrated strong capabilities in complex reasoning tasks such as mathematics (Cobbe et al., 2021) and programming (Jain et al., 2024). By incorporating chain-of-thought reasoning, these models perform multi-step logical deductions before answering, improving performance on challenging problems. This ability is often enhanced by reinforcement learning-based post-training, which equips models with self-reflection and error-correction skills (Gandhi et al., 2025) and enables exploration of multiple solution strategies. Research also shows that such reasoning mechanisms can be integrated into multimodal large models Wang et al. (2025) to enhance their effectiveness. As a result, the reasoning paradigm of large models is increasingly shifting toward deeper, "slow thinking"-based inference.

However, the "slow thinking" approach introduces a significant issue: "overthinking." Unlike traditional non-reasoning models, reasoning-based LRMs generate an analytical "chain of thought" (Wang et al., 2022) before producing a final answer. Recent studies indicate that even for simple questions, these models can produce reasoning chains that are significantly longer than responses from traditional models, while the final answer remains unchanged (Chen et al., 2024; Shen et al., 2025). This overthinking not only increases latency and exacerbates KV Cache pressure but also slows down the subsequent reinforcement learning fine-tuning process.

To enhance the reasoning efficiency and accuracy of large language models, mitigating overthinking has become a critical research direction. As a result, several recent studies have begun focusing on mitigating overthinking in reasoning models, such as through reinforcement learning methods based on length reward (Team et al., 2025; Luo et al., 2025; Aggarwal & Welleck, 2025; Shen et al., 2025) or efficient chain-of-thought supervised fine-tuning (SFT) strategies (Xia et al., 2025; Kang et al., 2025; Han et al., 2024). However, most existing approaches aim to encourage the model to produce

output that is as concise as possible, rather than guiding it to find a reasonable and adaptive response strategy. In fact, for certain complex problems, shorter answers are not necessarily better (Shen et al., 2025). Moreover, different models may exhibit distinct output behaviors across various datasets. Therefore, this research aims to explore a sustainable reinforcement learning training framework that enables models to adaptively adjust their output length range through long-term training, thereby effectively suppressing overthinking and supporting stable long-term optimization.

Based on the above discussion, this paper proposes a flexible online reinforcement learning method. Inspired by ideas from previous studies (Zhang et al., 2025; Shen et al., 2025), we estimate the difficulty of problems in real-time during online reinforcement learning based on the model's sampling results, and design a function to map problem difficulty to the expected response length. The difference between the expected length and the actual response length is then computed. To controllably tolerate deviations in length, we use a Gaussian function to map this difference into a length reward. After appropriate scaling, the length reward is added to the correctness reward to form the final reward signal.

During training, we observed that continuously applying the length reward could lead to excessive sensitivity of the length reward once the correctness reward begins to converge. When training steps are numerous, this may cause an excessive reduction in response length in later stages, ultimately impairing model performance. Therefore, we designed a reward trend monitor that uses an exponentially moving averaged correctness reward to track trends from short-term to long-term. Based on these trends, the length reward is dynamically enabled or disabled to ensure that the correctness reward can converge adequately in later phases, thereby safeguarding the final performance of the model. Experimental results demonstrate that our method effectively guides the model's output length distribution toward an optimum, thereby enabling the accuracy to converge more rapidly to a higher level. The contributions of this paper are summarized as follows:

- We propose a flexible function for predicting the desired output length and a corresponding mechanism for calculating length rewards.

- We design a reward trend monitor that automatically enables or disables the length reward based on an analysis of trends from short-term to long-term.

- Our approach adaptively converges to an appropriate output length range based on the model's inherent capability and the difficulty of the task, which in turn effectively promotes higher accuracy.

## 2 RELATED WORKS

### 2.1 LARGE REASONING MODELS AND THE PHENOMENON OF OVERTHINKING

Large reasoning models, by employing mechanisms such as Chain-of-Thought (CoT) and self-reflection that resemble human reasoning processes, can allocate more cognitive resources when tackling complex problems. This significantly enhances their capabilities in tasks like mathematical reasoning and program verification (Xu et al., 2025; Li et al., 2025b; Chen et al., 2025). OpenAI's O1 model (Jaech et al., 2024) highlighted that increasing the length of reasoning during the response generation process can markedly improve model performance, leading to the introduction of their reasoning model. Subsequently, other large reasoning models such as DeepSeek-R1 (Guo et al., 2025), Kimi (Team et al., 2025), and QWQ (Team, 2024) have been proposed. Among these, models following the R1 style—which encapsulate the reasoning process within special tokens like $< think >$ and $< /think >$—have become a benchmark paradigm for reasoning models in the open-source community. Although lengthy CoT reasoning significantly boosts accuracy, this step-by-step thinking mechanism also results in verbose output responses, consequently introducing substantial computational overhead and increased inference latency (Chen et al., 2024; Team et al., 2025). Furthermore, studies suggest that long contexts generated by over-thinking may increase the uncertainty and variance of outputs, potentially even leading to a decline in accuracy (Ghosal et al., 2025). Concurrently, over-thinking might make models more susceptible to malicious attacks, posing potential security risks (Kuo et al., 2025; Fang et al., 2025). Therefore, effectively guiding models to produce reasonable and efficient outputs has become a critical problem requiring urgent resolution.

## 2.2 REINFORCEMENT LEARNING-BASED INFERENCE LENGTH CONTROL

To reduce overthinking in deep reasoning models, a solution is to incorporate a reward for response brevity during reinforcement learning training. For instance, Training (Arora & Zanette, 2025) penalizes longer responses directly in the reward function to steer the model toward shorter outputs. ThinkPrune (Hou et al., 2025) sets a length threshold and assigns a zero reward to answers that exceed this threshold without solving the problem. O1-Pruner (Luo et al., 2025) pre-samples to estimate the baseline performance of the language model and subsequently employs an off-policy approach for training. L1 (Aggarwal & Welleck, 2025) introduces a fixed length budget into the reinforcement learning process and penalizes responses that exceed this budget. Recent studies focus on adaptive thinking length, which dynamically allocates reasoning effort based on problem difficulty. DAST (Shen et al., 2025) constructs length preference data according to question difficulty and length budget, and uses the SimPO method for training. ACPO (Cheng et al., 2025) estimates problem difficulty and length budget online to enable adaptive switching between fast and slow thinking modes. AALC (Li et al., 2025a) utilizes validation set rewards to dynamically adjust the weight of the length reward. Similar to prior work, this paper employs a difficulty-based length budget. However, we redesign the reward calculation and budget estimation functions, and introduce a distinct two-phase scheduling strategy: the length reward is emphasized early to accelerate convergence and de-emphasized later to prioritize accuracy.

## 3 METHOD

As illustrated in Figure 1, our method can be divided into three modules: first, an estimated length function is applied, followed by the calculation of a length reward based on the estimated length. Finally, a reward trend monitor performs real-time trend monitoring to dynamically adjust the length reward.

### 3.1 PROBLEM SETUP

This paper focuses on the research of "slow thinking" models with explicit reasoning capabilities in Large Language Models (LLMs), such as Chain-of-Thought (CoT) models. The aim is to optimize the balance between the conciseness and effectiveness of their reasoning processes. Specifically, we intend to minimize the length of intermediate reasoning steps (measured by the number of tokens) as much as possible without compromising the model's performance. This problem can be formulated as a constrained optimization problem: under the condition of ensuring that the model's prediction accuracy is close to its theoretical upper limit of capability, we seek a reasoning strategy that minimizes the length of intermediate reasoning.

Let $\mathcal{M}$ be a large language model with reasoning ability. For an input question x, it can generate an intermediate reasoning process r and finally output an answer a. The variables are defined as follows:

- $q \in \mathcal{Q}$: input question;
- $r \in \mathcal{R}$: The intermediate reasoning text generated by the model, with a length of $L(r)$ (counted by the number of tokens);
- $a \in \mathcal{A}$: The final answer output by the model;
- $y \in \mathcal{Y}$: The true answer to the question.

The reasoning ability of the model is influenced by the generated r, and the correctness of the final answer is represented by the indicator function $\mathbb{I}(a = y)$. We assume that there exists a theoretical upper limit of ability $P^*$, which is the maximum accuracy that the model can achieve under the optimal reasoning strategy:

$$P^* = \max_{r \sim \mathcal{M}} \mathbb{E}_{q,y}[\mathbb{I}(a = y)] \tag{1}$$

In the actual generation process, the model typically generates $r$ using various decoding strategies. Therefore, we introduce an inference strategy $\pi$ to control the generation method of $r$ (such as sampling temperature, maximum generation length, prompt strategy, etc.). Our goal is to find the optimal strategy $\pi^*$, which minimizes the average length of intermediate inference under the premise

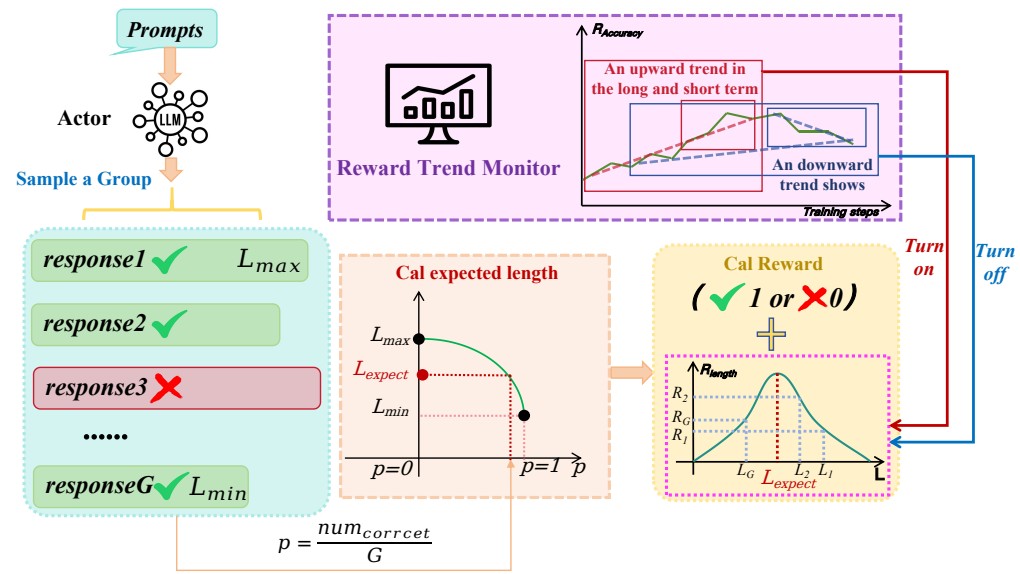

Figure 1: Our Method

that the accuracy rate is not lower than $P^* - \delta$ (where $\delta \geq 0$ represents the tolerable performance loss):

$$\pi^* = \arg\min_{\pi} \mathbb{E}_{q,r \sim \mathcal{M}_\pi}[L(r)]$$
$$\text{s.t. } \mathbb{E}_{q,y}[\mathbb{I}(a = y)] \geq P^* - \delta \tag{2}$$

To further formalize this problem, we define the following functions:

- $f(\pi) = \mathbb{E}_{q,y}[\mathbb{I}(a = y)]$: Accuracy under strategy $\pi$;

- $g(\pi) = \mathbb{E}_{q,r \sim \mathcal{M}_\pi}[L(r)]$: The average reasoning length under strategy $\pi$.

Then the optimization problem can be written as:

$$\min_{\pi} g(\pi) \quad \text{s.t.} \quad f(\pi) \geq P^* - \delta \tag{3}$$

Finally, we can also introduce a weighted objective function to unify accuracy and conciseness:

$$\mathcal{J}(\pi) = \lambda \cdot (1 - f(\pi)) + (1 - \lambda) \cdot \frac{g(\pi)}{T} \tag{4}$$

Here, T is the normalization factor (e.g., the maximum allowable growth length), and $\lambda \in [0, 1]$ is the weight coefficient. At this point, the problem is transformed into minimizing $\mathcal{J}(\pi)$.

## 3.2 LENGTH REWARD DESIGN

Our method is applicable to common large-model reinforcement learning algorithms (such as PPO (Schulman et al., 2017), GRPO (Shao et al., 2024) and DAPO (Yu et al., 2025)). This paper will take GRPO as an example to conduct method derivation and experiments.

As illustrated in the GRPO diagram, this approach abandons the typical critic model—which is usually the same size as the policy model—and instead estimates a baseline from group scores. Specifically, for each question $q$ sampled from the dataset distribution $P(Q)$, GRPO uses the old policy model $\pi_{\theta_{old}}$ to generate $G$ completions $\{o_1, o_2, \cdots, o_G\}$, Subsequently, GRPO optimizes the

policy model $\pi_\theta$ by maximizing the following objective:

$$\mathcal{I}_{GRPO}(\theta) = \mathrm{E}[q \sim P(Q), \{o_i\}_{i=1}^G \sim \pi_{\theta_{old}}(O|q)]$$

$$\frac{1}{G} \sum_{i=1}^G \left( \min \left( \frac{\pi_\theta(o_i|q)}{\pi_{\theta_{old}}(o_i|q)} A_i, \mathrm{clip} \left( \frac{\pi_\theta(o_i|q)}{\pi_{\theta_{old}}(o_i|q)}, 1-\varepsilon, 1+\varepsilon \right) A_i \right) - \beta \mathrm{D}_{KL} \left( \pi_\theta || \pi_{ref} \right) \right),$$

$$\mathbb{D}_{KL} \left( \pi_\theta || \pi_{ref} \right) = \frac{\pi_{ref}(o_i|q)}{\pi_\theta(o_i|q)} - \log \frac{\pi_{ref}(o_i|q)}{\pi_\theta(o_i|q)} - 1,$$

$$A_i = \frac{r_i - \max(\{r_1, r_2, \cdots, r_G\})}{\mathrm{std}(\{r_1, r_2, \cdots, r_G\})}.$$

where $\mathcal{E}$ and $\beta$ are hyperparameters, and $A_i$ is the advantage value, computed from the rewards of the outputs for each set of questions $\{r_1, r_2, \ldots, r_G\}$.

Due to the fact that the optimization objective represented by Formula 4 cannot be updated through gradient backpropagation, it can be designed as a reward function. Minimizing Formula 4 is equivalent to maximizing the following formula:

$$\mathcal{C}(\pi) = f(\pi) - \frac{1-\lambda}{\lambda T} g(\pi) \tag{5}$$

Let $\gamma = \frac{1-\lambda}{\lambda T}$, then Formula 5 can be written as:

$$\mathcal{C}(\pi) = f(\pi) - \gamma g(\pi) = \mathbb{E}_{q,r,a \sim \mathcal{M}_\pi} \left[ \mathbb{I}(a=y) - \gamma L(r) \right] \tag{6}$$

In practice, to estimate the functions $f(\pi)$ and $g(\pi)$ , we usually need a labeled dataset $D = \{(x_i, y_i)\}_{i=1}^N$, where N is the number of samples, $x_i$ is the input question, and $y_i$ is the corresponding ground truth answer. Then, for each sample, we run the model M under strategy $\pi$ to generate the reasoning text $r_i$ and the final answer $a_i$. Based on these outputs, we can calculate the estimated values $\hat{f}(\pi)$ and $\hat{g}(\pi)$:

$$\hat{f}(\pi) = \frac{1}{N} \sum_{i=1}^N \mathbb{I}(a_i = y_i), \hat{g}(\pi) = \frac{1}{N} \sum_{i=1}^N L(r_i) \tag{7}$$

Therefore, the optimization objective represented by Formula 6 can be written as:

$$\mathcal{C}(\pi) = \frac{1}{N} \sum_{i=1}^N \mathbb{I}(a_i = y_i) - \gamma \frac{1}{N} \sum_{i=1}^N L(r_i) \tag{8}$$

From Equation 8, it is intuitively thought that to maximize $\mathcal{C}(\pi)$, we can design the reward function as the following formula:

$$R(q_i, r_i, a_i) = \mathbb{I}(a_i = y_i) - L(r_i)$$

However, there are two problems with designing the reward function directly in this way: (1) The dimensions of $\mathbb{I}(a_i = y_i)$ and $L(r_i)$ are different, requiring balancing; (2) Our goal is not to blindly minimize the length, but rather to make the length close to a reasonable expected value $L_{\mathrm{expect}}$.Therefore, we introduce a length penalty term based on the Gaussian kernel to encourage the length generated by the model to be close to $L_{\mathrm{expect}}$, rather than blindly pursuing shortness. Meanwhile, the correctness reward is retained. Thus, the reward function is designed as follows:

$$R(q_i, r_i, a_i) = \mathbb{I}(a_i = y_i) + \exp(-\frac{(L - L_{expect})^2}{2 \left( \frac{L_{expect}}{k} \right)^2}) \tag{9}$$

As shown in Figure 2a, when the generated length deviates from the expected length, the reward will decrease. Here, K is a hyperparameter used to control the tolerance for length deviation.

Regarding $L_{\mathrm{expect}}$, we conduct real-time estimation during online learning based on the difficulty level of questions. In GRPO, for each question $q_i$, multiple answers $\{O_1, \ldots, O_G\}$ are sampled.

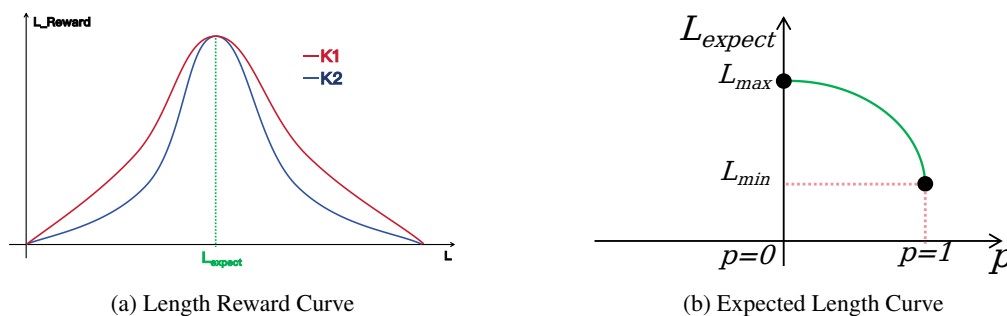

(a) Length Reward Curve  (b) Expected Length Curve

Figure 2: The function curve of length reward varying with length and the function curve of expected length varying with accuracy rate

Among these answers, let $C$ represent the number of correct ones, $p = C/G$ represent the correct rate, $L_{max}$ represent the length of the longest correct answer, and $L_{min}$ represent the length of the shortest correct answer. Then, we use the following formula to estimate the accuracy rate:

$$L_{expect} = p^2 L_{min} + (1 - p^2) L_{max} \tag{10}$$

As shown in Figure 2b, as the accuracy rate changes, the expected length also varies between $L_{max}$ and $L_{min}$. When the accuracy rate is close to 1, the expected length will be close to $L_{min}$; this is to explore the lower limit of the model's output length on the premise of ensuring accuracy. If the accuracy rate drops significantly, this function can ensure that the expected length increases rapidly, thereby guaranteeing the model's reasoning process. In practice, if the accuracy rate is 0, set $L_{max}$ to the maximum length of the model's output responses.

Finally, we add a proportional hyperparameter $\alpha$ to the length reward to control the scale of the length reward relative to the correctness reward, and multiply it by the accuracy rate $p$. When the accuracy rate is low, the impact of the length reward is reduced. Therefore, our final reward function is as follows:

$$R(q_i, r_i, a_i) = \mathbb{I}(a_i = y_i) + \alpha p \exp\left(-\frac{(L - L_{expect})^2}{2\left(\frac{L_{expect}}{k}\right)^2}\right) \tag{11}$$

### 3.3 REWARD TREND MONITOR

To ensure the convergence of the final correctness reward during the training process, we have designed a Reward Trend Monitor, which is used to track both the long-term and short-term trends of the reward curve. Based on these trends, it enables or disables the length reward. Specifically, we record the average value of the correctness reward at each step in the training history, denoted as $\{RC_1, \ldots, RC_n\}$, and then perform an Exponential Moving Average (EMA) smoothing on this sequence:

$$S_t = \beta \cdot RC_t + (1 - \beta) \cdot S_{t-1}$$

At each step, a linear fit is performed on the exponential moving averages of the short-term historical data and the long-term historical data to determine the reward growth trend. Let $w_1, \ldots, w_l$ represent the sizes of multiple monitoring windows ranging from short-term to long-term. Thus, we can obtain data for multiple trends: $\{S_{n-w_1}, \ldots, S_n\}, \ldots, \{S_{n-w_l}, \ldots, S(n)\}$. Perform linear fitting on the data of each window to get slopes $\{k_1, \ldots, k_l\}$ corresponding to the multiple windows; these slopes are used to represent the change trend of correctness reward from short-term to long-term. Set a hyperparameter $k_d$; when all values in $\{k_1, \ldots, k_l\}$ are less than $k_d$, the length reward is turned off:

$$(k_1 < k_d) \wedge \cdots \wedge (k_s < k_d) \implies R_{length} = 0$$

Table 1: A comparison of FOE-RL's performance on the MATH-TEST and MATH500 datasets, with bold and underlined values indicating the best results.

| Models | MATH-TEST | | | MATH500 | | |
|---|---|---|---|---|---|---|
| | Acc↑ | Tokens↓ | ACU↑ | Acc↑ | Tokens↓ | ACU↑ |
| Qwen3-0.6B-ORIGIN | 51.88 | 2308 | 3.75 | 49.60 | 2321 | 3.56 |
| Qwen3-0.6B-NL | 60.70 | 1897 | 5.33 | 63.00 | 1907 | 5.51 |
| Qwen3-0.6B-FOE | **1560** | **1560** | **6.55** | **63.40** | **1564** | **6.76** |
| Qwen3-1.7B-ORIGIN | 50.16 | 2606 | 1.13 | 53.00 | 2598 | 1.20 |
| Qwen3-1.7B-NL | 72.10 | 1976 | 2.14 | 73.40 | 1961 | 2.20 |
| Qwen3-1.7B-FOE | **76.10** | **1158** | **3.86** | **77.20** | **1138** | **3.99** |

Table 2: A comparison of FOE-RL's performance on the AMC23 , AIME2024 and AIME2025 datasets, with bold and underlined values indicating the best results.

| Models | AMC23 | | | AIME2024 | | | AIME2025 | | |
|---|---|---|---|---|---|---|---|---|---|
| | Acc↑ | Tokens↓ | ACU↑ | Acc↑ | Tokens↓ | ACU↑ | Acc↑ | Tokens↓ | ACU↑ |
| Qwen3-0.6B-ORIGIN | 27.50 | 2808 | 1.63 | 6.67 | 3051 | 0.36 | 6.67 | 3057 | 0.36 |
| Qwen3-0.6B-NL | 30.00 | 2556 | 1.96 | 6.67 | 3048 | 0.36 | **13.33** | 3031 | **0.73** |
| Qwen3-0.6B-FOE | **40.00** | **2232** | **2.99** | 6.67 | **3004** | **0.37** | 6.67 | **2986** | 0.37 |
| Qwen3-1.7B-ORIGIN | 25.00 | 2915 | 0.5 | 3.33 | 3072 | 0.06 | 6.67 | 3072 | 0.13 |
| Qwen3-1.7B-NL | **52.50** | 2514 | 1.23 | 13.33 | 3064 | 0.26 | 13.33 | 3006 | 0.26 |
| Qwen3-1.7B-FOE | 50.00 | **1866** | **1.58** | **26.67** | **2848** | **0.55** | **16.67** | **2690** | **0.36** |

# 4 EXPERIMENT

## 4.1 EXPERIMENTAL SETUP

**Datasets and Evaluation Metrics.** We trained the Qwen3-0.6B and Qwen1.7B (Team, 2025) models on the training set of the MATH (Hendrycks et al., 2021) dataset, which contains approximately 7.47K math problems. The models were then evaluated on the test set of MATH, comprising 5K samples. Additionally, performance was assessed on mathematical benchmarks including AIME24, AIME25, MATH500, and AMC23. During testing, we have the model generate one sample per question to calculate the average accuracy. We also record the average response length in tokens for each dataset, and employ the ACU (Ma et al., 2025) metric to evaluate the balance between accuracy and length:

$$\text{ACU} = \frac{\text{Accuracy}}{\#\text{Params} \times \#\text{Tokens}}$$

Due to computational constraints, both training and testing on the MATH dataset were conducted with a prompt length limited to 256 tokens and a response length capped at 3072 tokens. As a result, the actual amount of training data used was slightly smaller than the full dataset size, and truncation of prompts during testing may have led to some performance degradation. However, these limitations do not affect the validity of the conclusions drawn in this study.

**Experimental Details.** The original model (denoted by the suffix **-ORIGIN**) and the model trained only with correctness rewards (denoted by the suffix **-NL**) will be used as the baselines for comparison. During training, the model's sampling parameters were configured as follows: temperature = 1, top_k = -1 (disabled), and top_p = 1. During testing, the sampling settings were adjusted to temperature = 1, top_k = 50, and top_p = 0.7. The learning rate was 5e-7. In Equation 11, the

hyperparameter K for calculating the length reward was set to 3, and the scaling factor $\alpha$, which controls the weight of the length reward relative to the correctness reward, was set to 0.01. For the math dataset, the maximum prompt length and maximum response length during both training and testing were set to 256 and 3072, respectively. For all other datasets, the maximum prompt length and maximum response length during testing were configured as 1024 and 3072. The model was trained for one epoch on the MATH training set.

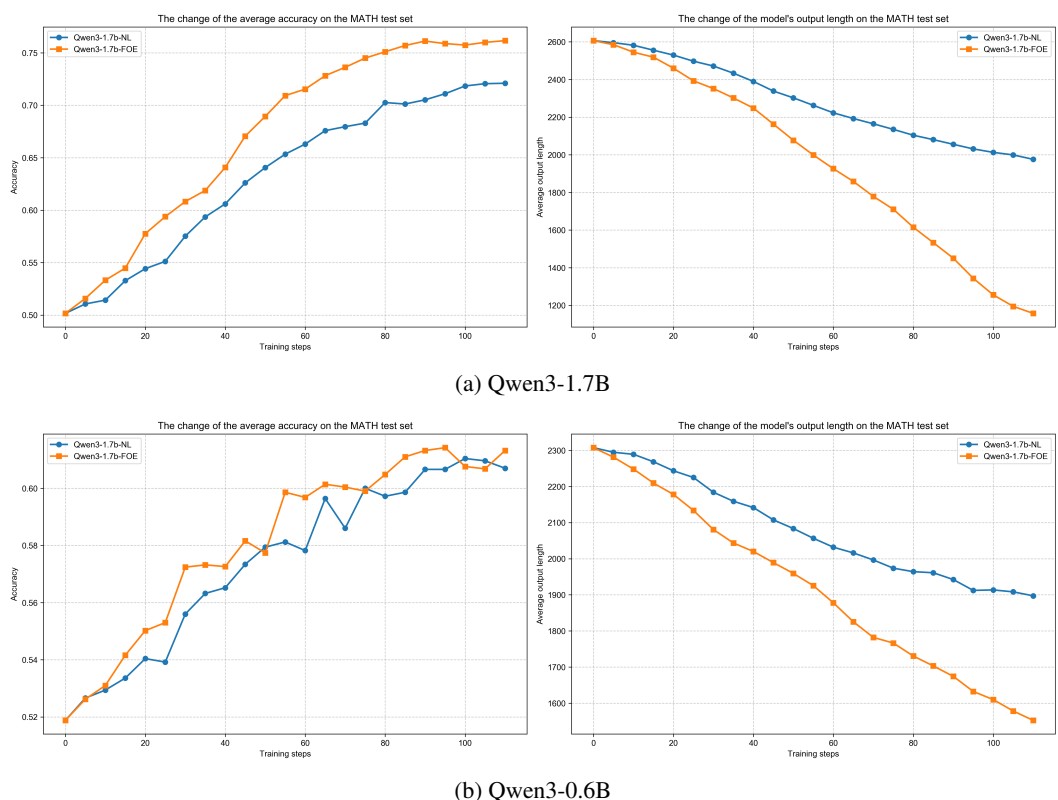

(a) Qwen3-1.7B

(b) Qwen3-0.6B

Figure 3: The model's accuracy and average response length on the MATH test set at different steps during training.

## 4.2 MAIN RESULTS ANALYSIS

Our main results are presented in Tables 1 and 2. Overall, our method achieves a reduction in token usage across all datasets while improving performance on most of them.

On the larger-scale MATH-TEST and MATH500 datasets, the Qwen3-1.7B model trained with our method achieves a reduction of over 40% in token usage alongside an accuracy improvement of more than 4%, compared to the model trained with only correctness rewards. The less capable Qwen3-0.6B model also demonstrates both reduced token consumption and improved performance under our method. This indicates that our approach can effectively identify an appropriate output length range suitable for the model's capability and problem difficulty, thereby enhancing model performance.

On the smaller-scale yet highly challenging datasets—AMC23, AIME24, and AIME25—our method consistently reduces token usage compared to models trained solely with correctness rewards, while also improving performance in most cases. For the weaker Qwen3-0.6B model, although significant performance gains are challenging on difficult, small-scale benchmarks like AIME2024 and AIME2025, token usage is still reduced. For the more powerful Qwen3-1.7B model, our method yields substantially greater token savings while maintaining or even matching the accuracy level.

Table 3: Comparison between the FOE Method and the Method with the Reward Trend Monitor Removed

| Models | MATH-TEST | | | MATH500 | | |
|---|---|---|---|---|---|---|
| | Acc↑ | Tokens↓ | ACU↑ | Acc↑ | Tokens↓ | ACU↑ |
| Qwen3-1.7B-OL | 46.74 | 268 | 10.26 | 48.60 | 258 | 11.08 |
| Qwen3-1.7B-FOE | 76.10 | 1158 | 3.86 | 77.20 | 1138 | 3.99 |

Furthermore, we observe that more capable models achieve greater token savings, and simpler datasets lead to more pronounced reductions. This robustly demonstrates that our method dynamically finds an optimal balance based on the model's capability and the problem's difficulty, effectively guiding the model to converge more quickly to its ideal output length range for different problems.

### 4.3 TRAINING ANALYSIS

As shown in Figure 3, we evaluated the accuracy of the Qwen3-0.6B and Qwen3-1.7B models—trained with the FOE method versus trained only with correctness rewards—on the MATH test set at intervals of every 5 training steps, while also recording the average length of generated responses at each step.

From Figure 3a, it can be observed that Qwen3-1.7B-FOE consistently achieved higher accuracy than Qwen3-1.7B-NL at every step. After 60 training steps, the performance gap stabilized at over 4 percentage points. Moreover, the response length of Qwen3-1.7B-FOE decreased significantly faster, dropping from above 2600 tokens to below 1100 within 110 steps.

In Figure 3b, due to the relatively weaker capability of Qwen3-0.6B, the accuracy of Qwen3-0.6B-FOE did not substantially surpass that of Qwen3-1.7B-NL, yet it remained higher in most training steps. Additionally, its response length decreased more rapidly, from an average of over 2300 to below 1600, whereas Qwen3-1.7B-NL only decreased to around 1900.

These experimental results demonstrate that more capable models exhibit faster convergence in output length. Furthermore, the proposed method enables accuracy to converge more quickly to a higher level, which benefits from its ability to guide the model's output length space toward an optimal state.

### 4.4 ABLATION ANALYSIS OF THE REWARD TREND MONITOR

We removed the length reward scheduling based on the reward monitor and trained the model (suffix -OL) by applying the length reward throughout the entire process. As shown in Table 3, although the output length was significantly reduced, the performance dropped to an unacceptable level, resulting in a performance degradation of over 30%. Furthermore, by analyzing the reward behavior during training, we found that this approach could not support long-term training effectively. For details, please refer to the appendix A.3.

## 5 CONCLUSION

We propose FOE-RL, a reinforcement learning method that effectively mitigates model "overthinking". Its core strength lies in guiding the model to converge rapidly to its optimal output length space—a range suited to its capability and the problem's difficulty. This adaptive length control is key to our method's success: by avoiding both excessively long and unnecessarily short reasoning paths, FOE-RL enables the model to achieve higher accuracy more efficiently. Experiments confirm that this approach not only significantly reduces token usage but, crucially, accelerates convergence to a superior level of performance . FOE-RL provides a principled solution for developing more efficient and capable reasoning models.

## ETHICS STATEMENT

This work adheres to the ICLR Code of Ethics. Our research focuses on improving the efficiency and accuracy of large reasoning models through reinforcement learning, without involving human subjects, sensitive data, or real-world deployment. All datasets used in this study (e.g., MATH, AIME, AMC) are publicly available and contain only mathematical problems with no personal or identifying information. Our methodology is designed to reduce computational overhead and inference latency, which aligns with the goal of promoting environmentally sustainable and accessible AI systems. We have conducted no user studies, and the models used (Qwen3-0.6B and Qwen3-1.7B) are publicly released under permissive licenses. We acknowledge that while our method encourages concise reasoning, it may still reflect biases present in the base models or training data. We encourage future work to assess the fairness and robustness of such efficiency-oriented training approaches in diverse contexts.

## REPRODUCIBILITY STATEMENT

To support reproducibility, we have provided detailed descriptions of our method, datasets, and experimental setup in Sections 3 and 4. Key hyperparameters (e.g., learning rate, temperature, K=3, $\alpha = 0.01$) are explicitly stated. The MATH, AIME, and AMC datasets are publicly accessible, and we specify the prompt and response length limits used during training and evaluation. The reward function design (Equation 11) and trend monitoring mechanism are fully described. The source code for implementing the FOE-RL framework, including the length reward scheduler and trend monitor, is included as an ancillary file with this submission and will be made publicly available upon acceptance. All model checkpoints used are based on publicly released Qwen3 models.

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

# A APPENDIX

## A.1 USE OF LLMS

In this study, large language models were employed solely for the linguistic polishing of certain text portions, with the aim of reducing grammatical errors and enhancing expression clarity, thereby aligning the writing more closely with the stylistic and terminological standards of scientific papers. It is important to note that the models were not utilized for any core research activities—including but not limited to research conception, literature search, data analysis, or conclusion formulation—so as to maintain the originality and academic integrity of the research process.

## A.2 ANALYSIS OF LENGTH REWARD SCHEDULING

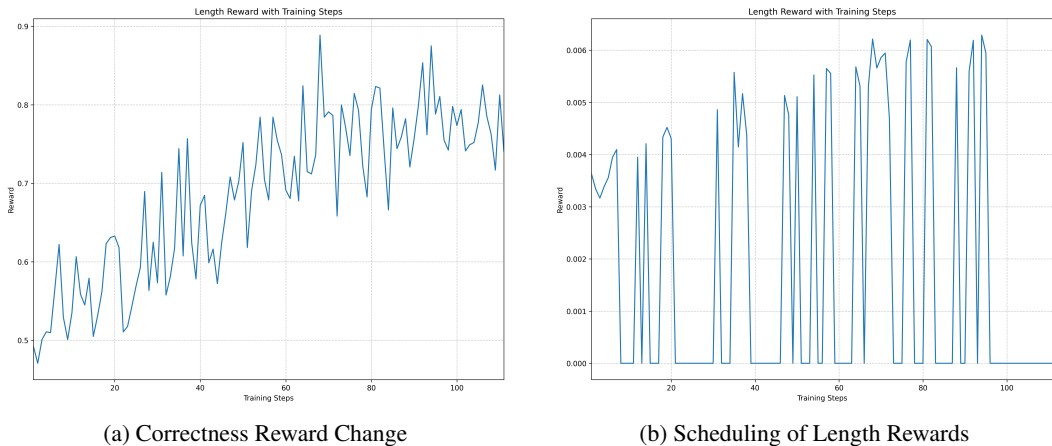

(a) Correctness Reward Change                     (b) Scheduling of Length Rewards

Figure 4: The change of correctness reward and scheduling of length rewards in Qwen3-1.7B-FOE during the training process

As shown in Figure 5, which illustrates the variation in correctness reward during the training process of the Qwen3-1.7B model and the deactivation of the length reward, it can be observed that the length reward is disabled whenever any long-term or short-term decline trend occurs in the correctness reward, thereby ensuring an improvement in the correctness reward. Additionally, the appropriate application of the length reward helps the model identify an optimal output length range.

## A.3 REWARD ANALYSIS WITHOUT LENGTH REWARD SCHEDULING

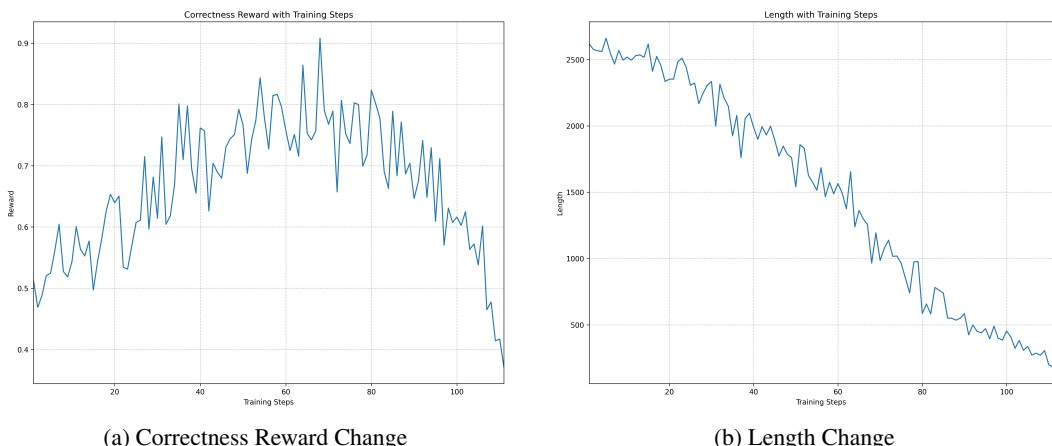

(a) Correctness Reward Change  (b) Length Change

Figure 5: The changes in correctness rewards and length changes with length rewards enabled throughout the training process

As shown in Figure 5a, when the length-based reward scheduling mechanism using the reward trend monitor is removed, the reward continues to rise in the first half of training but declines rapidly in later stages, significantly impairing model performance. Figure 5b indicates that this decline is caused by an excessive reduction in output length, which pushes the model beyond its optimal output length range. Moreover, around step 70, the model achieves the highest reward value, with an average output length of approximately 1000 tokens—consistent with the final output length of Qwen3-1.7B-FOE. This further underscores that our method effectively guides the model to converge to an optimal output length range.

