# OpenReview forum: "FOE-RL: Flexible Online Reinforcement Learning for Efficient Inference in Large Language Models"
_ICLR.cc/2026/Conference — Submitted to ICLR 2026_

### Official Review · Reviewer_ErGi · 2025-10-28

**Soundness:** 2
**Presentation:** 1
**Contribution:** 2
**Rating:** 4
**Confidence:** 3

**Summary:**

This paper addresses the "overthinking" problem in large reasoning models (LRMs), where models produce unnecessarily long chains of thought, increasing latency. The authors propose FOE-RL to adaptively encourage concise reasoning. FOE-RL reduces tokens (often >40%) and sometimes improves accuracy versus a correctness-only baseline.

**Strengths:**

1. Overthinking in LRMs is timely and well-motivated .
2. The Gaussian length reward around $L_{\text{expect}}$ is intuitive and easy to add to GRPO-style pipelines.
3. The paper correctly identifies a common failure mode of length-based rewards: the model's accuracy can collapse as it "over-optimizes" for shortness (reward hacking).
4. For Qwen3-1.7B on MATH-TEST, FOE-RL improves accuracy from 72.10 → 76.10 while cutting tokens 1976 → 1158; similar on MATH500 (73.40 → 77.20, 1961 → 1138) .

**Weaknesses:**

1. Defining $L_{\text{expect}}$ from within-batch correctness with the rule  $L_{\text{expect}}{=}p^2L_{\min} + (1{-}p^2)L_{\max}$ and determine $p$ based on correct rate is heuristic and could be brittle . No sensitivity analysis is provided (and also the choice of Gaussian kernel in reward term). The only ablation removes the scheduler (-OL); there is no study isolating $p$ scaling, the Gaussian vs. alternatives, or real-time difficulty estimation strategies .
2. The monitor relies on EMA and linear-fit slopes across windows, toggling the length reward when all slopes fall below a threshold $k_d$ , but the paper does not specify window sizes, $k_d$, or robustness to noise.
3. Some Experimental setups are questioning:
    - Single epoch, single seeds, and no variance/confidence intervals (common point but still needs to point out here).
    - Hyperparameters $k{=}3$, $\alpha{=}0.01$  are fixed without tuning studies.
    - The ACU metric is unconventional and mixes scale with architecture size, complicating cross-model interpretation; justification is brief (since I don’t see any scaling law on model size? Only math reasoning with two small models (0.6B/1.7B)).
    - The training/eval length caps (256/3072) may confound gains by truncating prompts/outputs .
4. no results on code, science QA or other important domain despite claims that the paradigm generalizes (If generalizes, please give more illustrations here).

**Questions:**

1. Justify the Gaussian length reward centered at $L_{\text{expect}}$ versus simpler Huber/piecewise penalties; provide sensitivity or AUC-style comparisons and discuss gradient shapes at both “too short” and “too long” extremes.
2. Precisely specify the EMA coefficient(s), window sizes, linear fit protocol (weighted vs. plain OLS), and the **threshold** $k_d$ used to toggle the length reward; currently these are under-specified.
3. The variance over random seeds for Table 1/2 results may be added.
4. If you have the computation resources, does FOE-RL transfer to non-math reasoning (code, GSM8K, scientific QA) or larger models (7B or more)?
5. More details on efficiency testing may be needed.

---

### Official Review · Reviewer_SW4q · 2025-10-30

**Soundness:** 2
**Presentation:** 2
**Contribution:** 2
**Rating:** 2
**Confidence:** 5

**Summary:**

This paper proposes FOE-RL, a method designed to mitigate “overthinking’’ in large reasoning models by adaptively controlling their output length during training. The approach estimates task difficulty in real time and predicts an appropriate response length, introducing a Gaussian-based length reward combined with the correctness reward. A reward trend monitor dynamically enables or disables the length reward based on smoothed reward trends to prevent over-shortening in later training stages. Experiments on several mathematical reasoning benchmarks show that FOE-RL reduces token usage while improving or maintaining accuracy, demonstrating faster convergence and adaptive optimization of reasoning length.

**Strengths:**

1. The paper is clearly written and easy to follow.

2. The experimental results, despite some evaluation concerns, still show certain improvements in efficiency and accuracy.

**Weaknesses:**

1. The proposed method is largely heuristic. Both the mapping between expected length and difficulty (Eq. 10) and the Gaussian-shaped length reward (Eq. 11) are ad-hoc choices without theoretical/empirical justification, and the reward trend monitor is similarly heuristic. These designs also introduce many hyperparameters, reducing the method’s general applicability.

2. The theoretical formulation in Section 3.1 aims to minimize the reasoning length L under an accuracy constraint, yet Section 3.2 shifts to aligning L with $L_{expect}$ instead of directly minimizing it. This disconnect weakens the consistency and clarity of the overall framework.

3. The experimental setup is limited: using only 0.6B and 1.7B models makes it difficult to draw strong conclusions. This limitation is partially understandable given computational constraints, but evaluating on AMC23, AIME24, and AIME25 with a single sample per question still weakens the credibility of the conclusions, especially for such small models.

4. The analysis focuses mainly on accuracy and token count. The paper lacks deeper quantitative or qualitative evidence showing that the proposed mechanism genuinely improves token budget allocation or reasoning efficiency beyond shorter outputs.

**Questions:**

1. A fundamental question: while it makes sense that smaller p (lower accuracy) should correspond to larger L, the reverse may not always hold—when a question is simple but inherently verbose, L may remain long even if p is high. How does the proposed mapping account for such cases?

2. Why must Equation (10) take a specific quadratic and concave form? Could other functional forms (e.g., linear or convex) work equally well? Is there any theoretical or empirical justification for this particular choice?

3. In line 474, the paper mentions that the proposed method “could not support long-term training effectively.” Could the authors clarify what this means? If the method becomes unstable or ineffective under extended training, how can it be considered a practical or scalable solution?

---

### Official Review · Reviewer_4FTW · 2025-11-01

**Soundness:** 2
**Presentation:** 3
**Contribution:** 2
**Rating:** 4
**Confidence:** 4

**Summary:**

The paper proposes a novel approach to optimize reinforcement learning for LLMs by dynamically estimating the ideal response length using real-time reward signals. This method effectively reduces response length and enables more efficient and effective inference on reasoning tasks.

**Strengths:**

* The proposed method is novel.
* The presentation and organization of the paper are clear and easy to follow.

**Weaknesses:**

The main weakness of this paper is that the limited experiments reduces the reliability of its conclusions. The proposed method introduces several hyperparameters, yet it is evaluated on only one training dataset, which affects its generalizability. Moreover, the performance of GRPO often fluctuates across different random seeds. Considering that the authors trained for only one epoch, the absence of error bars further limits the reliability of the experimental results.

**Questions:**

* Is temperature = 1 a commonly used setting in GRPO or related works?
* For models at the 1.5B scale, GRPO typically requires more than five epochs to converge. Could the authors provide results from additional training epochs to verify its convergence?

---

### Official Review · Reviewer_fSfS · 2025-11-03

**Soundness:** 1
**Presentation:** 1
**Contribution:** 1
**Rating:** 0
**Confidence:** 4

**Summary:**

The paper addresses the problem of overthinking when training large language models with rl on mathematical reasoning tasks. It proposes two techniques: a length-based reward and a reward trend monitor to mitigate this issue.

**Strengths:**

The focus on concise training is meaningful and relevant for improving efficiency in reinforcement learning–based fine-tuning of language models.

**Weaknesses:**

Using a length reward to penalize long responses is not a new idea. Moreover, the description of the reward trend monitor is unclear, making it difficult to understand how it functions in practice. The contribution and novelty appear limited, and the empirical experiments are not sufficiently strong to establish the paper’s effectiveness.

**Questions:**

1. DAPO (DAPO: An Open-Source LLM Reinforcement Learning System at Scale) also uses an overlength penalty to discourage overly long responses. What are the specific advantages of your design compared to theirs?
2. Could you clarify Section 3.3? Some definitions are unclear — for example, what are $RC_n$ and $S_t$? Also, why does the length reward become zero after setting $k_d$?
3. The experiments are too simple. In Table 2, the FOE method shows a much larger accuracy improvement over NL (26.67 vs. 13.33) on aime 24. This result seems inconsistent; I would expect similar accuracy but shorter responses for FOE, rather than such a large performance gap.

---

### Meta-Review · Area_Chair_dykN · 2025-12-13

**Summary:**

All reviewers recommended rejection, and there was no author rebuttal. Therefore, the paper is a clear rejection.

**Reviewer Concerns:**

There was no author rebuttal.

**Reviewer Scores:**

No changes due to the absence of author rebuttal.

---

### Decision · Program_Chairs · 2026-01-26

Reject